

# Liquid-liquid phase separation in organic particles containing one and two organic species: importance of the average O:C

Mijung Song[1], Suhan Ham[1], Ryan J. Andrews[2], Yuan You[2], Allan K. Bertram[2]*

[1] {Department of Earth and Environmental Sciences, Chonbuk National University, Jeollabuk-do, Republic of Korea}

[2] {Department of Chemistry, University of British Columbia, Vancouver, BC, V6T 1Z1, Canada}

Correspondence to: A. K. Bertram (bertram@chem.ubc.ca)

**Abstract**

Recently, experimental studies have shown that liquid-liquid phase separation (LLPS) can occur in organic particles free of inorganic salts. Most of these studies used organic particles consisting of secondary organic materials generated in environmental chambers. To gain additional insight into LLPS in organic particles free of inorganic salts, we studied LLPS in organic particles consisting of one and two commercially available organic species. For particles containing one organic species, three out of the six particle types investigated underwent LLPS. In these cases, LLPS was observed when the O:C was ≤ 0.44 and the RH was between ~97 and ~100%. The mechanism of phase separation was likely nucleation and growth. For particles containing two organic species, thirteen out of the fifteen particle types investigated underwent LLPS. In these cases, LLPS was observed when the O:C was ≤ 0.58 and mostly when the RH was between ~ 90 and ~ 100% RH. The mechanism of phase separation was likely spinodal decomposition. In almost all cases when LLPS was observed (for both one-component and two-component particles), the highest RH at which two liquids was observed was 100 ± 2.0%, which has important implications for the cloud condensation nuclei (CCN) properties of these particles. These combined results provide additional evidence that LLPS needs to be considered when predicting the CCN properties of organic particles in the atmosphere.





## 2 **1 Introduction**

Depending on location, organic materials comprise 20 – 80 % of the mass of submicrometer
particles in the atmosphere (Zhang et al., 2007; Jimenez et al., 2009). While the exact chemical
composition of this organic material is uncertain, measurements have shown that the oxygen-
to-carbon elemental ratio (O:C) of this organic material ranges from roughly 0.2 to 1.0 (Zhang
et al., 2007; Hallquist et al., 2009; Jimenez et al., 2009; Heald et al., 2010; Ng et al., 2010).
Important organic functional groups include carboxylic acids, alcohols, polyols, sugars,
aromatic compounds, amine groups, ethers, and esters (Decesari et al., 2006; Gilardoni et al.,
2009; Hallquist et al., 2009). Organic particles can affect the Earth's energy budget directly by
scattering and/or absorbing solar radiation, and indirectly by serving as nuclei for cloud
formation (Kanakidou et al., 2005; Hallquist et al., 2009; IPCC, 2013; Knopf et al., 2018). In
addition, they can affect air quality and human health (Jang et al., 2006; Baltensperger et al.,
2008) and provide a medium for multiphase reactions (George and Abbatt, 2010; Shiraiwa et
al., 2011; Abbatt et al., 2012; Houle et al., 2015; Reed et al., 2017).
To predict the role of organic particles in the atmosphere, information on their possible phase
transitions under atmospheric conditions is required (Hanel, 1976; Martin, 2000; Krieger et al.,
2012; You et al., 2014; Freedman, 2017). One possible phase transition that particles can
undergo as the relative humidity (RH) varies in the atmosphere is liquid-liquid phase separation
(LLPS) (Pankow, 2003; Marcolli et al., 2006; Ciobanu et al., 2009: Bertram et al., 2011;
Krieger et al., 2012; Song et al., 2012a; Zuend and Seinfeld, 2012; Veghte et al., 2013; You et
al., 2014; O'Brien et al., 2015). During the last several years, researchers have focused on
LLPS in particles containing both organic materials and inorganic salts. These studies have
shown that these mixed particles can undergo LLPS when the O:C of the organic species is less
than ~ 0.56 but not greater than 0.80 (Bertram et al., 2011; Krieger et al., 2012; Smith et al.,
2012; Song et al., 2012a; Schill and Tolbert, 2013; You et al., 2013; You et al., 2014). The effect
of particle size, temperature, organic functional groups, pH, and viscosity on LLPS in particles
containing organic materials and inorganic salts has also been explored (Krieger et al., 2012;
You et al., 2014; Losey et al., 2016; Freedman, 2017). The importance of LLPS in particles
containing organic materials and inorganic salts for gas-particle partitioning (Zuend et al., 2010;



Zuend and Seinfeld, 2012; Shiraiwa et al., 2013), hygroscopic properties (Hodas et al., 2015), optical properties (Fard et al., 2018), and cloud condensation nuclei (CCN) properties (Ovadnevaite et al., 2017) has also been investigated.

More recently, researchers have started to investigate LLPS in organic particles free of inorganic salts (Renbaum-Wolff et al., 2016; Rastak et al., 2017; Song et al., 2017). These studies have shown that LLPS can occur in particles containing secondary organic material (SOM) generated in environmental chambers when the O:C of the organic material is less than roughly 0.5. This work has also shown that LLPS occurs in SOM particles at RH values between approximately 95-100 % with important implications for the CCN properties of these particles (Petters et al. 2006; Hodas et al. 2016; Rastak et al., 2017; Renbaum-Wolff et al., 2016; Ovadnevaite et al., 2017).

Most of the previous experimental studies that investigated LLPS in organic particles free of inorganic salts focused on SOM particles generated in environmental chambers (Renbaum-Wolff et al., 2016; Rastak et al., 2017; Song et al., 2017). In the following, we studied LLPS in organic particles containing one and two commercially available organic species. Studies were carried out as a function of the average O:C of organic species to better constrain the O:C range required for LLPS. These studies provide additional insight into LLPS in organic particles free of inorganic salts, and the results from these studies should be useful for testing thermodynamic models used to predict LLPS in atmospheric particles.

## 2 Experiments

### 2.1 Materials

Listed in Table 1 are the commercially available organic species studied as well as the relevant properties of these organic species. The O:C of the organic species ranged from 0.29 to 0.75. All organic components were purchased from Sigma-Aldrich with purities ≥ 98 % and were used without further purification. In addition, all organic species studied are liquid at room temperature.

### 2.2 Particle production



Particles consisting of one organic species were generated by nebulizing the liquid organic
species without the addition of a solvent. Particles consisting of two organics species were
generated by first preparing a mixture of two liquid organics and then nebulizing the mixture,
again without the addition of a solvent. Based on visual observations, the mixtures of two liquid
organics studied were homogeneous (i.e. one phase) prior to nebulization. After nebulizing, the
generated organic particles were deposited onto siliconized glass slides (Hampton Research,
Canada). The nebulization and deposition process (followed by coagulation) resulted in organic
particles suspended on the glass slides with lateral dimensions ranging from ~30 to ~80 μm.
**2.3 Observations of LLPS using optical microscopy**
After deposing the organic particles on the glass slides, the glass slides were mounted in a
temperature and RH controlled flow-cell coupled to an optical microscope (Olympus BX43,
40× objective) (Parsons et al., 2004; Pant et al., 2006; Song et al., 2012a). The temperature of
the flow-cell was kept at $290 \pm 1$ K in all experiments. RH in the flow-cell was controlled by a
continuous flow of humidified $N_2$ gas. The total flow rate of the gas was fixed at ~500 sccm.
The RH was measured using a humidity and temperature sensor (Sensirion, Switzerland),
which was calibrated by observing the deliquescence RH of pure potassium carbonate (44 %
RH), sodium chloride (76 % RH), ammonium sulfate (80.5 % RH), and potassium nitrate (93.5 %
RH) particles (Winston and Bates, 1960). The uncertainty of the RH was $\pm$ 2.0 % after
calibration.
At the beginning of an experiment to probe LLPS, the organic particles were equilibrated at
~100 % RH for ~10 - 15 minutes. The RH was then reduced from ~100 % to ~ 0 %, followed
by an increase to ~100 % RH if no LLPS was detected. If LLPS was detected, the RH was
reduced from ~100 % to ~5 - 10 % lower than the RH at which the two liquid phases merged
into one phase, followed by an increase to ~100 % RH. The RH was adjusted at a rate of 0.1 -
0.5 % RH min$^{-1}$. During the experiments, optical images of several different particles were
recorded every 1 - 10 seconds using a video camera with a CMOS (Complementary Metal–
Oxide–Semiconductor) detector.

**3  Results and discussion**





### 3.1 Liquid-liquid phase separation in particles containing one organic species

At 290 ± 1 K, humidity cycles were performed with six different types of particles containing one organic species. Three out of the six particle types studied (particles containing diethyl sebacate, glyceryl tributyrate, and suberic acid monomethyl ester) underwent LLPS as RH was cycled from ~100 % to lower values and back to ~100 %. Shown in Fig. 1 and Movies S1 – S3 in the Supplement are images and movies as the RH was decreased for the three particles types that underwent LLPS. At ~100 % RH, diethyl sebacate particles (Fig. 1a) had a core-shell morphology. The outer phase was an organic-rich phase and the inner phase was a water-rich phase. At ~100 % RH, glyceryl tributyrate and suberic acid monomethyl ester particles (Fig. 1b-c) had inclusions of a water-rich phase suspended in an organic-rich phase. For all three particle types (Fig. 1a-c) as RH decreased from ~100 %, the amount of the water-rich phase decreased and eventually the two liquid phases were merged into one liquid phase at 99 - 97 % RH.

Figure 2 and Movies S4 – S6 in the Supplement show the same particles as those shown in Fig. 1 and Movies S1 - S3 but with increasing RH. In these cases, the particles remain as a single phase up to 98 - 99 % RH (Fig. 2). The light-colored circle in the center of each particle in the figure is due to the optical effect of light scattering (Bertram et al., 2011). Above ~ 99 % RH, two liquid phases are observed with the water-rich phase forming inclusions suspended in the organic-rich phase. As the RH increased the water-rich phase continued to grow. Based on the movies, the mechanism for formation of the water-rich phase is likely nucleation and growth (Movies S4 – S6) rather than spinodal decomposition.

In Table 2, the measured lower RH boundary for LLPS ($LLPS_{lower}$) and measured upper RH boundary for LLPS ($LLPS_{upper}$) are reported. $LLPS_{lower}$ and $LLPS_{upper}$ represent the lowest and highest RH at which two liquid phases were observed in the experiments. Table 2 shows that $LLPS_{lower}$ and $LLPS_{upper}$ were the same (within the uncertainties of the measurements) for both the increasing RH and decreasing RH experiments. This suggests that the kinetic barrier to LLPS in these experiments is small. According to the Gibbs phase rule, two liquid phases co-exist in equilibrium with the gas phase only at a single RH. Within experimental uncertainty, our results are consistent with this rule. Table 2 also illustrates that LLPS can occur (but not always) in organic particles containing one organic species when the O:C ratio is ≤ 0.44.

### 3.2 Liquid-liquid phase separation in particles containing two organic species



Using combinations of the organic species shown in Table 1, we studied LLPS in particles
containing two organic species. Of the fifteen particle types investigated, thirteen particle types
underwent LLPS during humidity cycles. Shown in Fig. 3 and Movies S7 - S9 (Supplement)
are images of three of the particle types that underwent LLPS (glyceryl tributyrate/polyethylene
glycol-400, diethyl sebacate/polyethylene glycol-400, and polyprophylene glycol/polyethylene
glycol-400) as RH decreased from ~100 %. The three particle types shown consisted of two
liquid phases over the largest RH range observed in our experiments. At ~ 100 % RH, glyceryl
tributyrate/polyethylene glycol-400 particles (Fig. 3a) and diethyl sebacate/polyethylene
glycol-400 particles (Fig. 3b) had a core-shell morphology with the shell consisting of an
organic-rich phase and the core consisting of a water-rich phase. As the RH decreased, the two
liquid phases merged into one liquid phase (Fig. 3a - b and Movies S7 – S8 in the Supplement).
The polypropylene glycerol/polyethylene glycol-400 particles (Fig. 3c) also underwent LLPS
as the RH decreased, but their behavior was slightly different. At ~ 100 % RH, only a single
phase was observed. As the RH decreased, a very thin outer shell was observed as well as
inclusions in the core of the particle at ~89 % RH. The shell and inclusions were most likely
an organic-rich phase. At ~74 % RH the two liquid phases merged into one (Fig. 3c and Movie
S9).
Figure 4 and Movies S10 – S12 (Supplement) correspond to the same particles as shown in Fig.
3 and Movies S7 - S9 but for increasing RH. At the lowest RH values studied, all three particle
types consisted of a single organic-rich phase (Fig. 4a-c). As the RH increased, numerous small
inclusions appeared throughout the particles. These small inclusions then coalesced resulting
in a core-shell morphology. Based on the movies (Movies S10 - S12), the mechanism of LLPS
was likely spinodal decomposition, a phase transition that occurs without an energy barrier
(Ciobanu et al., 2009; Song et al. 2013). LLPS in SOM particles produced from the ozonolysis
of α-pinene, limonene, and β-caryophyllene also appeared to occur by this mechanism
(Renbaum-Wolff et al., 2016; Song et al., 2017).
In Table 3 the measured $LLPS_{lower}$ and $LLPS_{upper}$ for particles containing two organics are
reported as a function of the average O:C in the particles. Only results from the increasing RH
experiments are shown. Within uncertainty, the same results were obtained for decreasing RH
(Table S1 in the Supplement), indicating a kinetic barrier to the LLPS in these experiments was
small. The average O:C values for particles containing two organic species were calculated





using Eq. S1 in the Supplement. The only particles containing two organics that did not undergo
LLPS were particles consisting of propylene glycol/diethyl L-tartrate and polyethylene glycol-
400/diethyl L-tartrate (Table 3). These two particles had the highest and third highest average
O:C values of all the two component particles investigated (Table 3). This is consistent with
reported phase transitions in bulk solutions containing two organics and water (Ganbavale et
al., 2015). To illustrate, in a bulk mixture of two organics and water with a low average O:C
value (e.g. a mixture of 1-butanol, 1-propanol, and water) two liquids can form (Ganbavale et
al., 2015). On the other hand, in a bulk mixture of two organics and water with a high average
O:C value (e.g. a mixture of acetic acid, ethanol, and water), only a single liquid is observed
(Ganbavale et al., 2015) .
Table 3 illustrates that LLPS in particles containing two organics was mostly observed between
~ 90 and ~ 100 % RH. The exception to this was particles containing polyethylene glycol-400
mixed with glyceryl tributyrate, diethyl sebacate, or polyprophylene glycol. In these three cases
two liquid phases were observed over a much wider range, and in one case from ~5 % to ~100 %
RH. Table 3 also illustrates that when LLPS occurred in particles containing two organic
species, $LLPS_{upper}$ was $100 \pm 2\%$ in almost all cases. This is important as it suggests that these
organic particles can have an organic-rich shell at high RH, which could lower the surface
tension and influence the CCN properties of the particles (Petters et al. 2006; Hodas et al. 2016;
Renbaum-Wolff et al., 2016; Rastak et al., 2017; Ovadnevaite et al., 2017).
**3.3 Comparison between particles containing one organic species, two organic species,**
**and secondary organic materials**
Shown in Fig. 5 are the lower and upper RH boundaries for LLPS ($LLPS_{lower}$ and $LLPS_{upper}$)
as a function of the O:C for particles containing one organic species (Fig. 5a), particles
containing two organic species (Fig. 5b), and particles consisting of SOM generated in
environmental chambers (Fig. 5c). Results for particles consisting of SOM were taken from
Renbaum-Wolff et al. (2016), Rastak et al. (2017), and Song et al. (2017). The O:C range of
the SOM particles is based on previous studies (Heaton et al., 2007; Lambe et al., 2015; Li et
al., 2015; Renbaum-Wolff et al., 2016; Rastak et al., 2017; Song et al., 2017).
Figure 5 suggests that LLPS in two-component organic particles (Fig. 5b) occurs over a wider
range of average O:C values than LLPS in one-component organic particles (Fig. 5a).   In


addition, LLPS in two-component organic particles occurs over a wider range of RH values
than LLPS in one-component organic particles. This illustrates that as the complexity of the
organic particles increases, LLPS can occur over a wider range of atmospheric conditions.
Figure 5 also illustrates that there is a relationship between the average O:C and the occurrence
of LLPS in organic particles. For particles containing one organic species, LLPS was observed
when the O:C was ≤ 0.44; for particles containing two organic species, LLPS was observed
when the O:C was ≤ 0.58; and for particles containing SOM, LLPS was observed when the
O:C was ≲ 0.44. In Fig. 5, the O:C range founded in ambient organic aerosols is also shown
(Zhang et al., 2007; Hallquist et al., 2009; Jimenez et al., 2009; Heald et al., 2010; Ng et al.,
2010). Based on this range, and the range over which LLPS was observed in one-component
organic particles, two-component organic particles, and SOM particles, LLPS is expected to
be a common feature of organic aerosols in the atmosphere.
Figure 5 also illustrates that where LLPS was observed, LLPS$_{upper}$ was 100 ± 2.0% in almost
all cases. In these cases, an organic-rich phase can form a shell around the particles and lower
the surface tension of the particles at high RH, with important consequences for the CCN
properties of the organic particles (Petters et al. 2006; Hodas et al. 2016; Renbaum-Wolff et al.,
2016; Rastak et al., 2017; Ovadnevaite et al., 2017). These combined results provide additional
evidence that LLPS needs to be considered when predicting the CCN properties of organic
particles in the atmosphere.

## 4   Summary and conclusions

We investigated LLPS at 290 ± 1 K in organic particles containing one and two organic species
free of inorganic salts. For organic particles containing one organic species, three of the six
different particle types studied underwent LLPS. For the three cases where LLPS was observed,
particles separated when the O:C was ≤ 0.44 and the RH was close to ~ 100 %. The mechanism
for LLPS in these particles was likely nucleation and growth. Using combinations of the
organic species, we also investigated LLPS in particles containing two organic species.
Thirteen out of the fifteen particle types investigated underwent LLPS, and LLPS was observed
when the O:C was ≤ 0.58. The mechanism of phase separation was likely spinodal



decomposition. In these cases, LLPS was observed between ~ 90 and ~ 100 % RH, except for three cases in which LLPS was observed over a much wider range of RH (5 % to 100 %). The combined results suggest that as the complexity of the organic particles increase, LLPS can occur over a wider range of atmospheric conditions. Since the O:C of organic particles in the atmosphere ranges from ~0.2 to ~1.0, these results also provide additional support for the suggestion that LLPS can occur in organic particles in the atmosphere.

**Acknowledgements**

This work was supported by the Natural Sciences and Engineering Research Council of Canada. Support from the National Research Foundation of Korea (NRF) grant funded by the Korea Government (MSIP) (2016R1C1B1009243) is also acknowledged.

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



1    Table 1. Organic species studied, as well as molecular formula, molecular weight, oxygen-to-

2    carbon elemental ratios (O:C), and functional groups of the species studied. All organic species

3    are liquid at room temperature.

| Compound | Molecular formula | Molecular weight (g mol$^{-1}$) | O:C | Functional group |
|---|---|---|---|---|
| Diethyl sebacate | $C_{14}H_{26}O_4$ | 258 | 0.29 | Ester |
| Poly(propylene glycol) | $C_{3n}H_{6n+2}O_{n+1}$ | 425 | 0.38 | Alcohol, ether |
| Glyceryl tributyrate | $C_{15}H_{26}O_6$ | 302 | 0.40 | Ester |
| Suberic acid monomethyl ester | $C_9H_{16}O_4$ | 188 | 0.44 | Carboxylic acid, ester |
| Polyethylene glycol-400 | $C_{2n}H_{4n+2}O_{n+1}$ | 400 | 0.56 | Alcohol, ether |
| Diethyl L-tartrate | $C_8H_{14}O_6$ | 206 | 0.75 | Carboxylic acid, alcohol, ether, ester |



Table 2. Lower RH boundary for LLPS (LLPS$_{lower}$) and upper RH boundary for LLPS
(LLPS$_{upper}$) for particles containing one organic species. LLPS$_{lower}$ and LLPS$_{upper}$ represent to
the lowest and highest RH at which two liquid phases were observed in the experiments.
Uncertainties represent 2σ of multiple measurements and the uncertainty from the calibration.
"No LLPS" indicates that only one phase was observed for the full range of relative humidity
explored (~ 0 to 100 % RH).

| Compounds | Increasing RH | | Decreasing RH | |
|---|---|---|---|---|
| | LLPS$_{lower}$ | LLPS$_{upper}$ | LLPS$_{lower}$ | LLPS$_{upper}$ |
| Diethyl sebacate (O:C=0.29) | 99.0 ± 2.8 | 100 ± 2.0 | 97.7 ± 3.1 | 100 ± 2.0 |
| Poly(propylene glycol) (O:C= 0.38) | No LLPS | | No LLPS | |
| Glyceryl tributyrate (O:C=0.40) | 98.8 ± 2.2 | 100 ± 2.0 | 97.9 ± 2.3 | 100 ± 2.0 |
| Suberic acid monomethyl ester (O:C=0.44) | 99.2 ± 2.1 | 100 ± 2.0 | 99.2 ± 2.2 | 100 ± 2.0 |
| Polyethylene glycol-400 (O:C=0.56) | No LLPS | | No LLPS | |
| Diethyl L-tartrate (O:C=0.75) | No LLPS | | No LLPS | |



Table 3. Lower RH boundary for LLPS (LLPS$_{lower}$) and upper RH boundary for LLPS (LLPS$_{upper}$) for particles containing one (yellow shading) and two organic species (grey shading). LLPS$_{lower}$ and LLPS$_{upper}$ represent the lowest and highest RH at which two liquid phases were observed in the experiments. Uncertainties represent $2\sigma$ of multiple measurements and the uncertainty from the calibration. "No LLPS" indicates that only one phase was observed for the full range of relative humidity (RH) explored (~ 0 to 100 % RH). Only results are shown here for increasing RH. For particles containing one organic species (yellow shading) the O:C of the organic is indicated in brackets. For particles containing two organics (grey shading) the average O:C of the particles is indicated in brackets.

| | Diethyl sebacate | Propylene glycol | Glyceryl tributyrate, | Suberic acid monomethyl ester | Polyethylene glycol-400 | Diethyl L-tartrate |
|---|---|---|---|---|---|---|
| Diethyl sebacate | LLPS$_{lower}$= 99.0 ± 2.8%, LLPS$_{upper}$= 100 ± 2.0% (O:C=0.29) | | | | | |
| Propylene glycol | LLPS$_{lower}$= 92.4 ± 2.1 %, LLPS$_{upper}$= 100 ± 2.0% (O:C=0.33) | No LLPS (O:C=0.38) | | | | |
| Glyceryl tributyrate | LLPS$_{lower}$= 96.3 ± 4.3 %, LLPS$_{upper}$= 100 ± 2.0% (O:C=0.34) | LLPS$_{lower}$= 93.8 ± 2.3 %, LLPS$_{upper}$= 100 ± 2.0% (O:C=0.39) | LLPS$_{lower}$= 98.9 ± 2.2 %, LLPS$_{upper}$= 100 ± 2.0% (O:C=0.40) | | | |
| Suberic acid monomethyl ester | LLPS$_{lower}$= 97.4 ± 3.1 %, LLPS$_{upper}$= 100 ± 2.0% (O:C=0.36) | LLPS$_{lower}$= 97.7 ± 2.3 %, LLPS$_{upper}$= 100 ± 2.0% (O:C=0.42) | LLPS$_{lower}$= 97.6 ± 3.4 %, LLPS$_{upper}$= 100 ± 2.0% (O:C=0.42) | LLPS$_{lower}$= 99.2 ± 2.1 %, LLPS$_{upper}$= 100 ± 2.0% (O:C=0.44) | | |

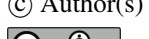



| | | | | | | |
|---|---|---|---|---|---|---|
| Polyethylene glycol-400 | LLPS$_{lower}$= 5.2 ± 3.8 %, LLPS$_{upper}$= 100 ± 2.0% (O:C=0.39) | LLPS$_{lower}$= 73.9± 2.5 %, LLPS$_{upper}$= 89.9 ± 3.0 % (O:C=0.47) | LLPS$_{lower}$= 16.0 ± 2.3 %, LLPS$_{upper}$= 100 ± 2.0% (O:C=0.47) | LLPS$_{lower}$= 93.2 ± 3.2 %, LLPS$_{upper}$= 100 ± 2.0% (O:C=0.50) | No LLPS (O:C=0.56) | |
| Diethyl L-tartrate | LLPS$_{lower}$= 92.8 ± 3.4 %, LLPS$_{upper}$= 100 ± 2.0% (O:C=0.48) | No LLPS (O:C=0.56) | LLPS$_{lower}$= 93.9 ± 3.3 %, LLPS$_{upper}$= 100 ± 2.0% (O:C=0.55) | LLPS$_{lower}$= 97.3 ± 2.7 %, LLPS$_{upper}$= 100 ± 2.0% (O:C=0.58) | No LLPS (O:C=0.68) | No LLPS (O:C=0.75) |



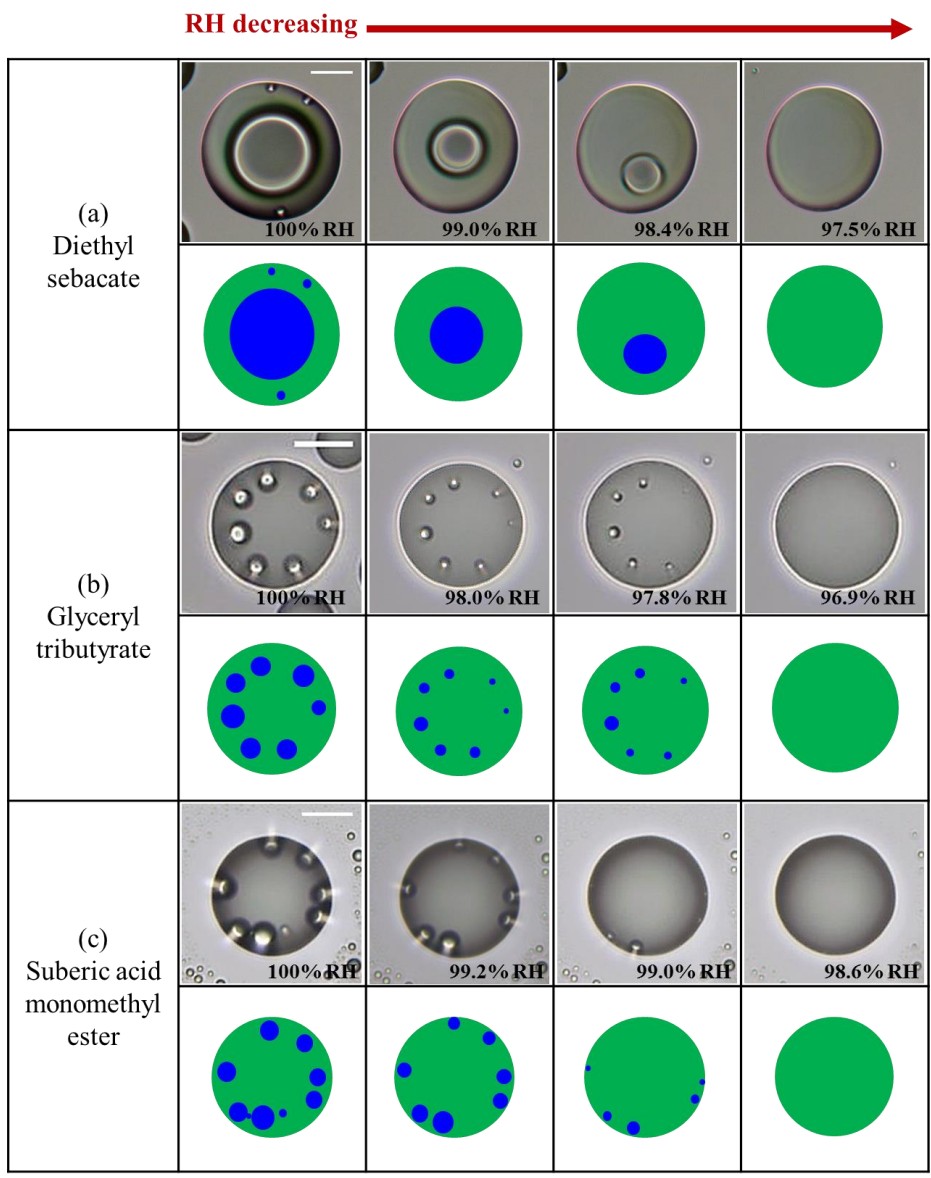

Figure 1. Optical images of single particles consisting of one organic species with decreasing RH: (a) diethyl sebacate particles, (b) glyceryl tributyrate particles, and (c) suberic acid monomethyl ester particles. Illustrations are presented below the images. Green: organic-rich phase. Blue: water-rich phase. The scale bar is 20 μm.



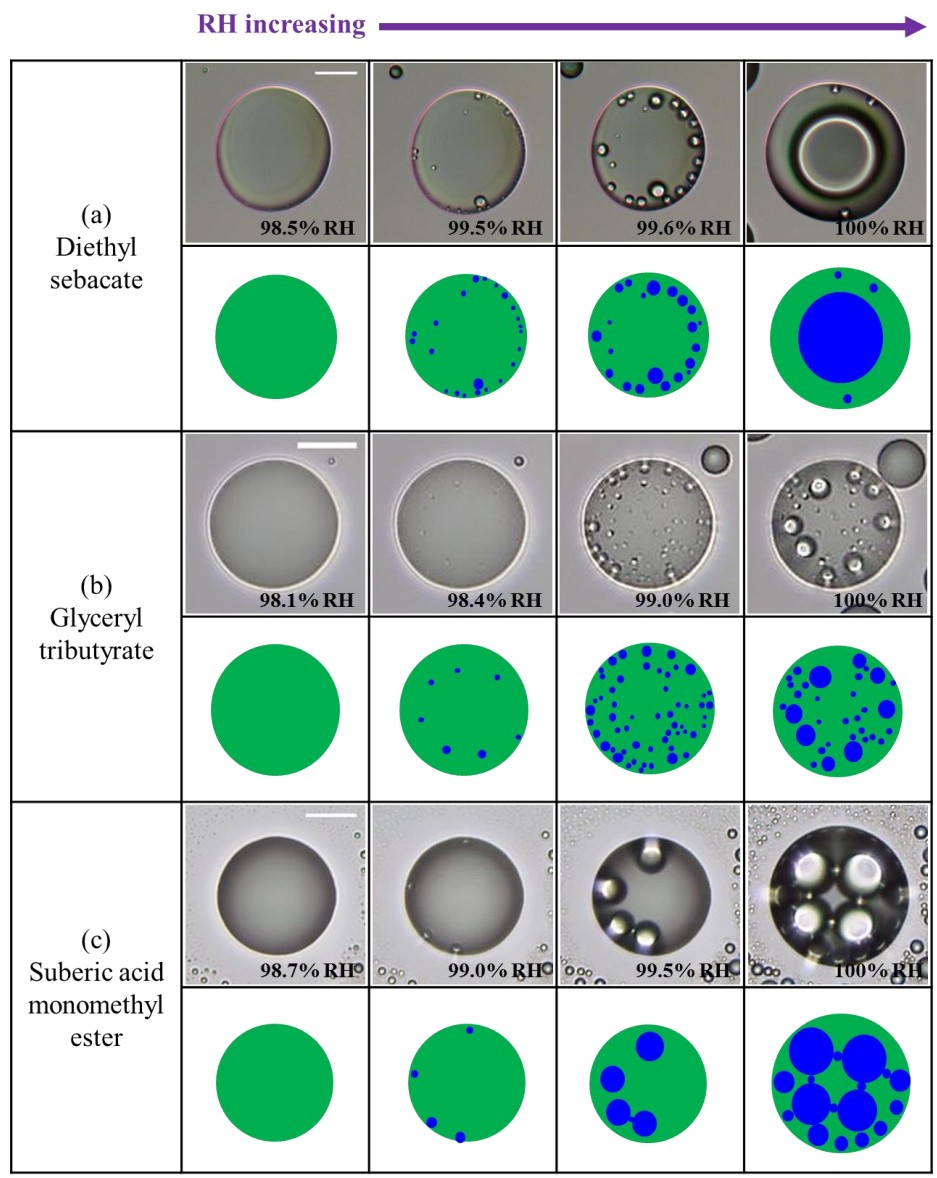

Figure 2. Optical images of single particles consisting of one organic species with increasing
RH: (a) diethyl sebacate particles, (b) glyceryl tributyrate particles, and (c) suberic acid
monomethyl ester particles. The particles are the same ones shown in Fig. 1, but for increasing
RH. Illustrations are presented below the images. Green: organic-rich phase. Blue: water-rich





phase. The scale bar is 20 µm.



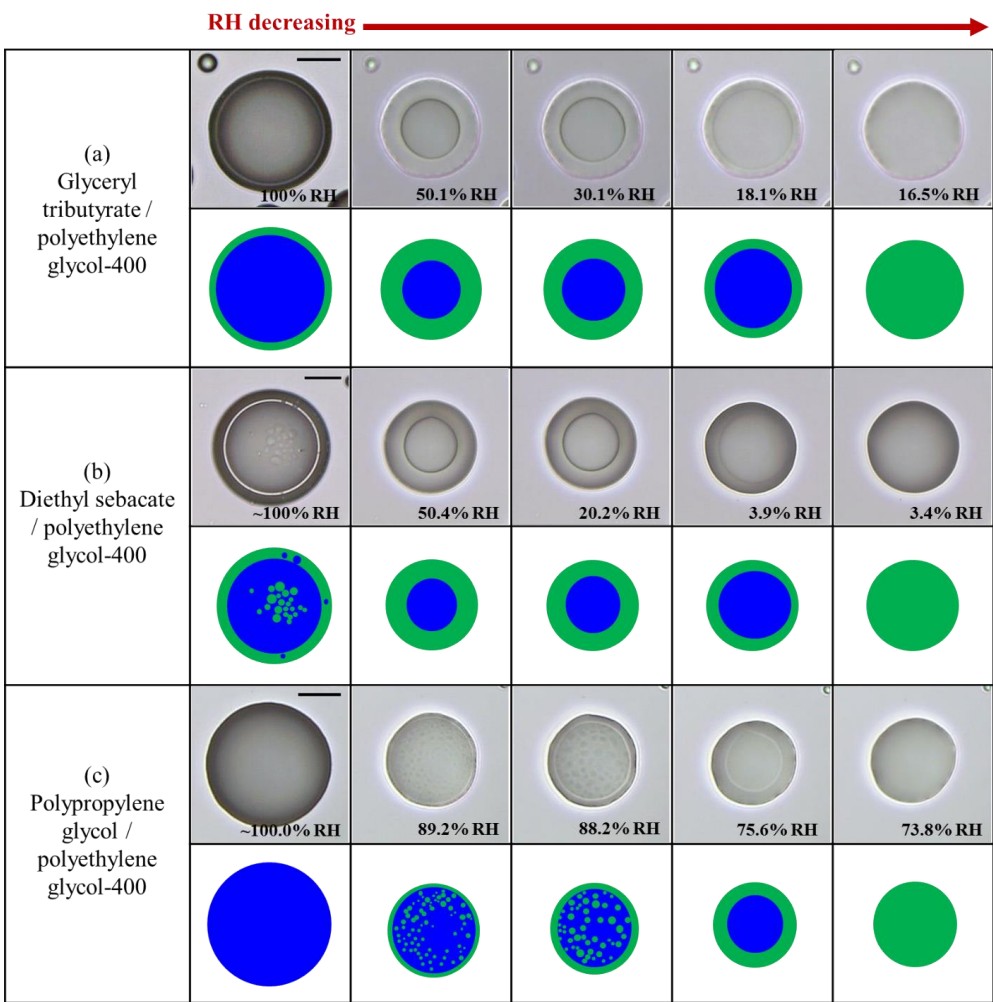

Figure 3. Optical images for particles consisting of two organic species for decreasing RH: (a)
glyceryl tributyrate/polyethylene glycol-400, (b) diethyl sebacate/ polyethylene glycol-400,
and (c) poly(propylene glycol)/polyethylglycol-400. Illustrations are presented below the
images for clarity. Green: organic-rich phase. Blue: water-rich phase. The scale bar is 20 µm.





Figure 4. Optical images for particles consisting of two organic species for increasing RH: (a) glyceryl tributyrate/polyethylene glycol-400, (b) diethyl sebacate/ polyethylene glycol-400, and (c) poly(propylene glycol)/polyethylglglycol-400. The particles are the same as shown in Fig. 3 but for increasing RH. Illustrations are presented below the images for clarity. Green: organic-rich phase. Blue: water-rich phase. The scale bar is 20 µm.





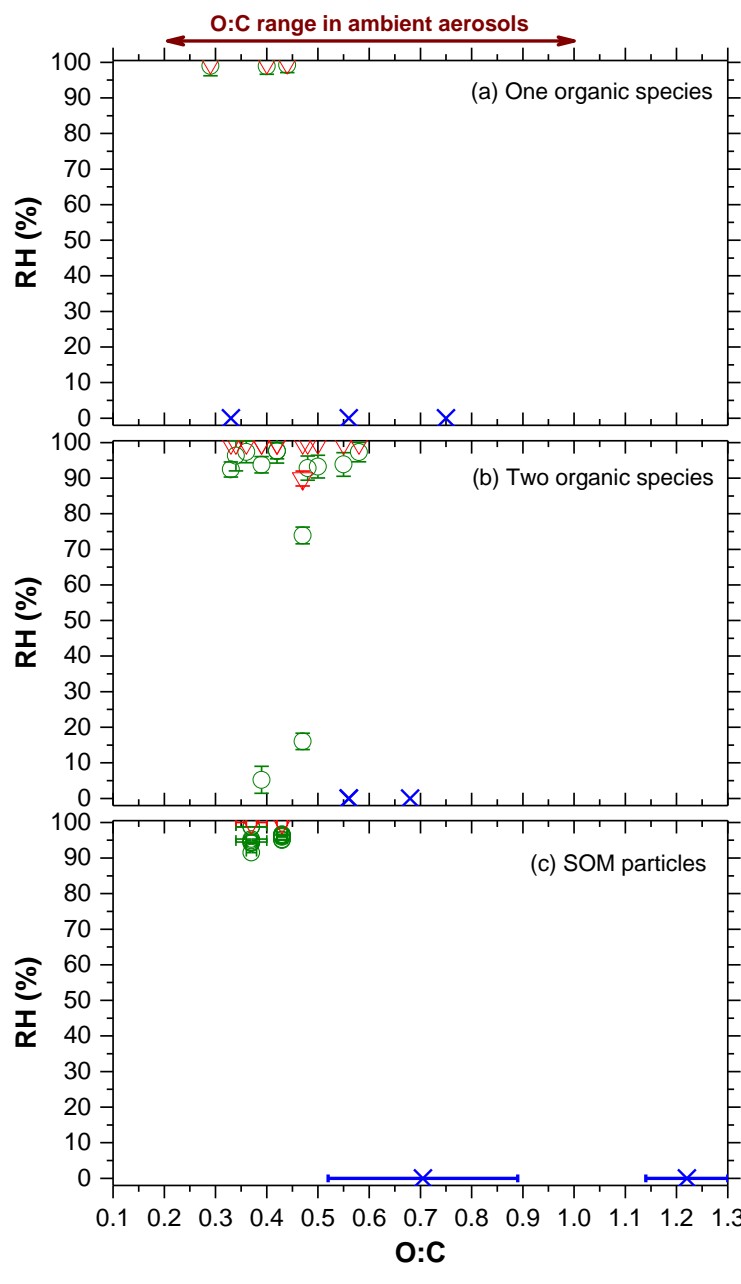

3    Figure 5. Lower and upper RH boundary for LLPS as a function of the O:C of the organic



particle: (a) particles containing one organic species, (b) particles containing two organic
species, and (c) particles consisting of secondary organic material generated in environmental
chambers. Only results for experiments with increasing RH are shown. Green circles represent
lower boundary for the LLPS, and red triangles represent upper boundary for LLPS. Blue
crosses indicate that LLPS was not observed during RH cycles. Uncertainties represent 2σ of
multiple measurements and the uncertainty from the calibration. Results for particles consisting
of secondary organic material (Panel c) were taken from Renbaum-Wolff et al. (2016), Rastak
et al. (2017), and Song et al. (2017). The O:C range of the SOM particles is based on previous
studies (Heaton et al., 2007; Lambe et al., 2015; Li et al., 2015; Renbaum-Wolff et al., 2016;
Rastak et al., 2017; Song et al., 2017). The arrow at the top of the figure represents the O:C
range of organic particles in the atmosphere (Zhang et al., 2007; Hallquist et al., 2009; Jimenez
et al., 2009; Heald et al., 2010; Ng et al., 2010).

