# Peer review of "Liquid-liquid phase separation in organic particles containing one and"

_Atmospheric Chemistry and Physics, 2018_

## Referee Comment (RC1) · Anonymous Referee #1 · 29 May 2018

I have reviewed the paper "Liquid-liquid phase separation in organic particles containing one and two organic species: importance of the average O:C" by M. Song et al. which was submitted to Atmospheric Chemistry and Physics Discussions. This paper builds nicely on the group's previous work, which showed that aerosol particles composed of secondary organic material can undergo liquid-liquid phase separation (LLPS) at high relative humidities. This phenomenon has been proposed to be important in the formation of CCN. This paper takes a more fundamental approach, showing that LLPS is found for much simpler systems consisting of 1-2 organic species. The paper is well written and concise, and its points are straightforward. It will make a good contribution to ACP after revisions. My two major comments are that 1) I do not think

[Figure]

that the authors can be so definitive about the O:C ratio needed for phase separation since they have used a limited number of systems and see exceptions to the ratio they state, and 2) the authors miss an opportunity to hypothesize about the fundamental chemistry that underlies the phenomenon that they observe.

Table 1: It would be helpful to have the molecular structure of these species listed as well. Please also include solubility.

Table 3: It would be helpful to know the mass fractions of the compounds in the mixture rather than having to back calculate based on the O:C ratio.

pg 5 line 5: Can you comment on the "stems" that seem to connect the inclusions to the edge of the particles in Figs. 1b and 1c? I assume these are an optical artifact.

pg 5 line 8: How do you know that the organic-rich phase is the outer phase? Are you basing this on surface tensions or was it measured? I have the same question for pg 6 lines 10 and 15.

pg 5 line 14: Ciobanu et al. 2009 make a distinction between nucleation and growth from the edge of the particle vs. nucleation and growth that originates in the interior of the particle. It looks like Movies S4 and S6 show nucleation and growth that originates at the edge, and in Movie S5, the growth originates from the interior of the particle. If this result is consistent across many particles, it is worth pointing out to the reader.

pg 5 line 22: Was there any size dependence to the range of LLPS values that is observed across a number of particles?

pg 5 line 30: I don't think you can draw any conclusion about the O:C ratios at which LLPS is observed based on Table 2 because some systems phase separate and others do not at O:C < 0.44 and there are not a lot of systems studied. Can you make any conclusions based on the functional groups?

Movie S7 was not included in the zip file I downloaded. Movies S8-S12 were not readable by Quicktime (even though they have the same file extension as the other

movies).

pg 6 line 23: Optical microscopy may not provide the best observation of the spinodal decomposition mechanism. Phase diagrams indicate that the binodal and spinodal meet at a single point for systems like PEG 400/ammonium sulfate/water. If the system has the exact composition that is needed to go through this critical point, the system will undergo LLPS by spinodal decomposition. At other compositions, it depends on the activation energy. Since you have hypothesized that the activation barrier is low based on the fact that the mixing and separation RH are the same or very similar, nucleation and growth is more likely. Ciobanu et al. 2009 report that they observe spinodal decomposition for a large range of systems, but Altaf et al. Chem Comm 2016, 52, 9220 shows for the same system that only one composition has no activation barrier (i.e. undergoes spinodal decomposition).

pg 7 line 14: PEG 400 mixed with diethyl sebacate or glyceryl tributyrate needs further explanation. Why are the ranges so large and the lower value of LLPS so low?

pg 7 line 29: Please acknowledge that the first two sentences are based on very small numbers of compounds.

pg 8 lines 4-9: Based on the fact that small numbers of compounds were used and there are exceptions in this range, the determining O:C ratios can be hypothesized, but should not be stated as definitively as in this paragraph.

Based on some of my comments above, the title to the paper is a bit strong, as I do not think the average O:C at which phase separation occurs can be concluded from this study, only hypothesized.

The authors miss an opportunity to comment on the fundamental chemistry that drives the behavior they observe. It seems odd that we can make solutions of these compounds, and yet they phase separate in droplets as more water is added to the system. Why don't they stay dissolved? On a similar note, if you added ammonium sulfate
to these solutions, would the same phenomenon be observed or would ammonium sulfate increase the solubility of these compounds at high RH (salting in behavior)?

---

## Referee Comment (RC2) · Anonymous Referee #2 · 31 May 2018

The paper investigates the liquid-liquid phase separation (LLPS) in organic particles. LLPS was observed when the O:C ratios of the particles were ≤ 0.44 for single organic species and ≤ 0.58 for mixtures containing two organic compounds. Most of the LLPS can be observed at RH close to 100%. The authors demonstrate that the LLPS may have important implications on the CCN properties of organic particles. The manuscript is well written, and the results are clearly presented. I would like to recommend this manuscript to be published in *Atmos. Chem. Phys.* if my following concerns are fully addressed.

1.  Page 4, Line 8

    The dimensions of the particles were measured around 30-80 μm. Depending on the particle generation methods, sizes of the particles used in the LLPS studies in the literature vary from 1 to 80 μm. 80 μm used in this study is at the higher end. Does particle size affect the results of LLPS?

2.  Page 5, Line 29

    a.  Have the authors tried nonanedioic acid (C9H16O4) and tetradecanedioic acid (C14H26O4)? These are the isomers of suberic acid monomethyl ester and diethyl sebacate with the same O:C ratio.

        One may expect different results with these isomers. For example, due to the stronger polarity and hydrogen bonding of acids as compared to esters, the acid compounds may have stronger interactions with water molecules, leading to higher SRHs. Or no LLPS may be observed. These effects may actually be helpful to explain why the authors did not observe LLPS with poly (propylene glycol) and PEG-400. Can you comment on whether/how the intermolecular interactions between organic compounds and water affect LLPS?

    b.  In contrast to the experiments with organic + ammonium sulfate conducted in the same group (e.g., Bertram et al.[1] and You et al.[2]), no LLPS was observed in this study using the same organic compounds (i.e., polyethylene glycol, PEG-400, and diethyl-L-tartrate). I wonder if the authors have any explanation regarding the different observations between these two studies.

3.  Page 6, Line 2 and Table 3

    What is the molar/mass ratio between the two organic compounds in the mixtures in Table 1? Is it 1:1? Have you tried any other mixing ratios? When nebulizing the mixture, how can the authors be sure that the composition of individual particle was the same as its original mixture?

4.  Page 6, Line 6

a. It is interesting that adding PEG-400 which itself did not give LLPS, expanded the LLPS RH range of other compounds. Does it simply due to the change of O:C ratio? If not, please explain it in more detail.

b. For polypropylene glycerol/PEG-400, no LLPS was observed in the single organic species scenarios. But mixing them together yielded LLPS with a low SRH (~75%). Have the authors tried any other mixtures of the following: diethyl sebacate, glyceryl tributyrate and suberic acid monomethyl ester? I wonder if the SRH and LLPS RH range will also change? Can you comment on this?

5. Page 8, Line 7

SOM is a bulk mixture. Why the results of SOM are similar to those from single organic components (i.e., LLPS occurred with O:C ratio ≤ 0.44; narrower LLPS RH range was observed)? Is it just a coincidence? It seems that O:C ratio solely cannot fully explain the results. Can the authors comment on this?

**Minor comments**

1. Page 4, Line 10: "deposing". Should it be "depositing"?

2. Page 17, Table 1: For diethyl-L-tartrate, I don't think there are ether group and carboxylic acid group.

**Reference**

(1) Bertram, A. K.; Martin, S. T.; Hanna, S. J.; Smith, M. L.; Bodsworth, A.; Chen, Q.; Kuwata, M.; Liu, A.; You, Y.; Zorn, S. R. Predicting the relative humidities of liquid-liquid phase separation, efflorescence, and deliquescence of mixed particles of ammonium sulfate, organic material, and water using the organic-to-sulfate mass ratio of the particle and the oxygen-to-carbon elemental ratio of the organic component. *Atmos. Chem. Phys.* **2011**, *11* (21), 10995–11006.

(2) You, Y.; Renbaum-Wolff, L.; Bertram, A. K. Liquid-liquid phase separation in particles containing organics mixed with ammonium sulfate, ammonium bisulfate, ammonium nitrate or sodium chloride. *Atmos. Chem. Phys.* **2013**, *13* (23), 11723–11734.

---

## Author Comment (AC1) · 6 Aug 2018

Summary: I have reviewed the paper "Liquid-liquid phase separation in organic particles containing one and two organic species: importance of the average O:C" by M. Song et al. which was submitted to Atmospheric Chemistry and Physics Discussions. This paper builds nicely on the group's previous work, which showed that aerosol particles composed of secondary organic material can undergo liquid-liquid phase separation (LLPS) at high relative humidities. This phenomenon has been proposed to be important in the formation of CCN. This paper takes a more fundamental approach, showing that LLPS is found for much simpler systems consisting of 1-2 organic species.

[Figure]

The paper is well written and concise, and its points are straightforward. It will make a good contribution to ACP after revisions.

Concerns. [1] My two major comments are that 1) I do not think that the authors can be so definitive about the O:C ratio needed for phase separation since they have used a limited number of systems and see exceptions to the ratio they state. and 2) the authors miss an opportunity to hypothesize about the fundamental chemistry that underlies the phenomenon that they observe.

[A1]To address the referee's comments we will 1) make it clearer in the revised manuscript that the average O:C is not the only parameter needed to predict LLPS (Abstract and page 6 line 7 – 8) and 2) include some discussion on the possible fundamental chemistry that underlies the phenomenon (page 8, line 15 - 28). Specifically, the following text will be added to the manuscript.

"Figure 5 also suggests that the average O:C is an important factor (although not the only factor) in determining the occurrence of LLPS in organic particles. For particles containing one organic species, LLPS was observed when the O:C was $\leq$ 0.44 (but not in all cases); for particles containing two organic species, LLPS was observed when the O:C was $\leq$ 0.58 (but not in all cases); and for particles containing SOM, LLPS was observed when the O:C was âĽŠ0.44. In Fig. 5, the O:C range founded in ambient organic aerosols is also shown (Zhang et al., 2007; Hallquist et al., 2009; Jimenez et al., 2009; Heald et al., 2010; Ng et al., 2010). Based on this range and the range over which LLPS was observed in one-component organic particles, two-component organic particles, and SOM particles, LLPS is likely a common feature of organic aerosols in the atmosphere. In addition to the average O:C and the types of functional groups, the spread in the O:C values of the organic molecules with the same particles is likely important for LLPS. Additional studies are needed to isolate the effect of average O:C, functional groups, and spread in O:C values on LLPS in organic particles."

[2] Table 1: It would be helpful to have the molecular structure of these species listed as well. Please also include solubility.

[A2] As suggested, we will add molecular structure and solubility to Table 1 of the revised manuscript.

[3] Table 3: It would be helpful to know the mass fractions of the compounds in the mixture rather than having to back calculate based on the O:C ratio.

[A3]All experiments for two organic species were conducted with the mass ratio of 1:1. This information will be added to Sect. 2.2 of the revised manuscript.

[4] pg 5 line 5: Can you comment on the "stems" that seem to connect the inclusions to the edge of the particles in Figs. 1b and 1c? I assume these are an optical artifact.

[A4]Yes, it is an optical artifact in Figs. 1b and 1c. In the revised manuscript this will be mentioned in the figure caption for clarity.

[5] pg 5 line 8: How do you know that the organic-rich phase is the outer phase? Are you basing this on surface tensions or was it measured? I have the same question for pg 6 lines 10 and 15.

[A5]We assume that the core of the particle is water-rich because the size of the core decreases and eventually disappears as the RH decreases. The surface tension of water and the surface tensions of organics are consistent with this assumption (Jasper, 1972). To address the referee's comment, we will add this information to the revised manuscript (Sect 3.1).

Jasper, J. J.: The surface tension of pure liquid compounds, J. Phys. And Chem. Ref. Data, vol 1, 841-1009, Doi: http://dx.doi.org/10.1063/1.3253106, 1972.

[6] pg 5 line 14: Ciobanu et al. 2009 make a distinction between nucleation and growth from the edge of the particle vs. nucleation and growth that originates in the interior of the particle. It looks like Movies S4 and S6 show nucleation and growth that originates

at the edge, and in Movie S5, the growth originates from the interior of the particle. If this result is consistent across many particles, it is worth pointing out to the reader.

[A6]Thank you for the comment. Nucleation and growth begun from the edge of the particles of diethyl sebacate (Movie S4) and suberic monomethyl ester (Movie S6) while nucleation started from the interior of the particles of glyceryl tributyrate (Movie S5). Such process was observed in all particles. This information will be added to the Sect. 3.1 of the revised manuscript.

[7] pg 5 line 22: Was there any size dependence to the range of LLPS values that is observed across a number of particles?

[A7]We did not observe a dependence of LLPS on the particle size, although only a small range of sizes were explored. This information will be added to the Sect. 3.1 of the revised manuscript.

[8] pg 5 line 30: I don't think you can draw any conclusion about the O:C ratios at which LLPS is observed based on Table 2 because some systems phase separate and others do not at O:C < 0.44 and there are not a lot of systems studied. Can you make any conclusions based on the functional groups?

[A8]To address the referee's comment, we will modify the sentence on page 6 line 7 - 8 to the following: "Table 2 also illustrates that LLPS can occur (but not always) in organic particles containing one organic species when the O:C ratio is $\leq$ 0.44." In addition, we will also add the following discussion: "In addition to the O:C ratio, the types of organic functional groups present in the molecules are also likely important for LLPS (Song et al., 2012b), since different functional groups lead to different strengths of intermolecular forces with water." The following reference will also be added: "Song, M., Marcolli, C., Krieger, U. K., Zuend, A., and Peter, T.: Liquid-liquid phase separation in aerosol particles: Dependence on O:C, organic functionalities, and compositional complexity, Geophysical Research Letters, 39, 10.1029/2012gl052807, 2012."

[Figure]

[9] Movie S7 was not included in the zip file I downloaded. Movies S8-S12 were not readable by Quicktime (even though they have the same file extension as the other to these solutions, would the same phenomenon be observed or would ammonium sulfate increase the solubility of these compounds at high RH (salting in behavior)?

[A9]For the issue of the movies, we will check with Copernicus Publication regarding readability and will add movie S7. Regarding LLPS process when ammonium sulfate is added, the presence of ammonium sulfate would likely enhance LLPS process due to salting out effect (Marcolli and Krieger, 2006) but further studies are needed to verify this point. Marcolli, C., and Krieger, U. K.: Phase changes during hygroscopic cycles of mixed organic/inorganic model systems of tropospheric aerosols, J Phys Chem A, 110, 1881-1893, Doi 10.1021/Jp0556759, 2006.

---

## Author Comment (AC2) · 6 Aug 2018

Summary: The paper investigates the liquid-liquid phase separation (LLPS) in organic particles. LLPS was observed when the O:C ratios of the particles were $\leq$ 0.44 for single organic species and $\leq$ 0.58 for mixtures containing two organic compounds. Most of the LLPS can be observed at RH close to 100%. The authors demonstrate that the LLPS may have important implications on the CCN properties of organic particles. The manuscript is well written, and the results are clearly presented. I would like to recommend this manuscript to be published in Atmos. Chem. Phys. If my following concerns are fully addressed.

[Figure]

Concern. [10] 1. Page 4, Line 8. The dimensions of the particles were measured around 30-80 $\mu$m. Depending on the particle generation methods, sizes of the particles used in the LLPS studies in the literature vary from 1 to 80 micrometers. 80 micrometer used in this study is at the higher end. Does particle size affect the results of LLPS?

[A10] The resolution of the microscope used in the current experiments was roughly 1 micrometer. From experience, detection of LLPS with our microscope setup is the clearest when the size of the particles are roughly 30 - 80 ïA∎m, although smaller sizes are possible with optical microscopy. We did not observe a size dependence for the LLPS, although only a narrow range of sizes were investigated. This information will be added to Sect. 3.1 of the revised manuscript.

[11] 2. Page 5, Line 29. a. Have the authors tried nonanedioic acid (C9H16O4) and tetradecanedioic acid (C14H26O4)? These are the isomers of suberic acid monomethyl ester and diethyl sebacate with the same O:C ratio. One may expect different results with these isomers. For example, due to the stronger polarity and hydrogen bonding of acids as compared to esters, the acid compounds may have stronger interactions with water molecules, leading to higher SRHs. Or no LLPS may be observed. These effects may actually be helpful to explain why the authors did not observe LLPS with poly (propylene glycol) and PEG-400. Can you comment on whether/how the intermolecular interactions between organic compounds and water affect LLPS?

[A11] Thank you for the comment. Nonanedioic acid and tetradecanedioic acid are solid at room temperature. In our studies we only used liquid organics species to ensure the mixtures were homogeneous prior to nebulization. Regarding intermolecular interactions, see response to comment [8] above.

[12] b. In contrast to the experiments with organic + ammonium sulfate conducted in the same group (e.g., Bertram et al.1 and You et al.2), no LLPS was observed in this study using the same organic compounds (i.e., polyethylene glycol, PEG-400, and diethyl-L-tartrate). I wonder if the authors have any explanation regarding the different

observations between these two studies.

[A12] The difference observations between the two studies is likely due to salting out by ammonium sulfate (Marcolli and Krieger, 2006). We are currently working on a manuscript that will focus on this point. Marcolli, C., and Krieger, U. K.: Phase changes during hygroscopic cycles of mixed organic/inorganic model systems of tropospheric aerosols, J Phys Chem A, 110, 1881-1893, Doi 10.1021/Jp0556759, 2006.

[13] 3. Page 6, Line 2 and Table 3 What is the molar/mass ratio between the two organic compounds in the mixtures in Table 1? Is it 1:1? Have you tried any other mixing ratios? When nebulizing the mixture, how can the authors be sure that the composition of individual particle was the same as its original mixture?

[A13] All experiments for two organic species were conducted with the mass ratio of 1:1. This information will be added to the Sect. 2.2 of the revised manuscript. All organic species used in this study are liquid at room temperature. Based on visual observations, the mixtures of two liquid organics were one phase prior to nebulizing as described in Sect. 2.2, hence a change on composition of the individual particles was not expected due to nebulizing the mixture. This information will be added to the revised manuscript.

[14] 4. Page 6, Line. a. It is interesting that adding PEG-400 which itself did not give LLPS, expanded the LLPS RH range of other compounds. Does it simply due to the change of O:C ratio? If not, please explain it in more detail.

[A14] In addition to the average O:C, the spread in O:C values within an organic particle and the types of organic functional groups are likely also important. The manuscript will be modified to make these points clear (Sections 3.1).

[15] b. For polypropylene glycerol/PEG-400, no LLPS was observed in the single organic species scenarios. But mixing them together yielded LLPS with a low SRH (∼75%). Have the authors tried any other mixtures of the following: diethyl sebacate,

glyceryl tributyrate and suberic acid monomethyl ester? I wonder if the SRH and LLPS RH range will also change? Can you comment on this?

[A15]We are not sure if we completely understand the referee's comments. The following binary mixtures related to the referee's comments were studied: diethyl sebacate + glyceryl tributyrate, diethyl sebacate + suberic acid monomethyl ester, and glyceryl tributyrate + suberic acid monomethyl ester (see Table 3). Is the referee suggesting we try ternary mixtures? We have not tried any ternary mixtures.

[16] 5. Page 8, Line 7. SOM is a bulk mixture. Why the results of SOM are similar to those from single organic components (i.e., LLPS occurred with O:C ratio $\leq$ 0.44; narrower LLPS RH range was observed)? Is it just a coincidence? It seems that O:C ratio solely cannot fully explain the results. Can the authors comment on this?

[A16] First, the uncertainty of O:C of the SOM mixtures are large and thus we cannot determine based on our results if the SOM results are in better agreement with the studies using one organic species or two organic species. Second, we agree that the O:C ratio cannot solely explain the results. The manuscript will be revised to try and make this point clearer (Section 3.3).

Minor comments [17] 1. Page 4, Line 10: "deposing". Should it be "depositing"?

[A17] We will correct "deposing" to "depositing" in the revised manuscript.

[18] 2. Page 17, Table 1: For diethyl-L-tartrate, I don't think there are ether group and carboxylic acid group.

[A18] Thank you for the correction. Diethyl-L-tartrate is alcohol and ester group. We will correct it in Table 1 of the revised manuscript.

Reference (1) Bertram, A. K.; Martin, S. T.; Hanna, S. J.; Smith, M. L.; Bodsworth, A.; Chen, Q.; Kuwata, M.; Liu, A.; You, Y.; Zorn, S. R. Predicting the relative humidities of liquidliquid phase separation, efflorescence, and deliquescence of mixed particles of ammonium sulfate, organic material, and water using the organic-to-sulfate mass ratio

of the particle and the oxygen-to-carbon elemental ratio of the organic component. Atmos. Chem. Phys. 2011, 11 (21), 10995–11006. (2) You, Y.; Renbaum-Wolff, L.; Bertram, A. K. Liquid-liquid phase separation in particles containing organics mixed with ammonium sulfate, ammonium bisulfate, ammonium nitrate or sodium chloride. Atmos. Chem. Phys. 2013, 13 (23), 11723–11734.

---

## Author Response (AR2)

Prof. Ryan Sullivan

Co-Editor of Atmospheric Chemistry and Physics

Dear Ryan,

Listed below are our responses to the comments from the referees of our manuscript. For clarity and visual distinction, the referee comments or questions are listed here in black and are preceded by bracketed, italicized numbers (e.g. *[1]*). Author's responses are offset in red below each referee statement. We thank the referees for carefully reading our manuscript and for their helpful comments!

Sincerely,

Allan Bertram,

Professor, Department of Chemistry

University of British Columbia

**Response to Referee #1**

Summary:

I have reviewed the paper "Liquid-liquid phase separation in organic particles containing one and two organic species: importance of the average O:C" by M. Song et al. which was submitted to Atmospheric Chemistry and Physics Discussions. This paper builds nicely on the group's previous work, which showed that aerosol particles composed of secondary organic material can undergo liquid-liquid phase separation (LLPS) at high relative humidities. This phenomenon has been proposed to be important in the formation of CCN. This paper takes a more fundamental approach, showing that LLPS is found for much simpler systems consisting of 1-2 organic species. The paper is well written and concise, and its points are straightforward. It will make a good contribution to ACP after revisions.

Concerns.
*[1]* My two major comments are that 1) I do not think that the authors can be so definitive about the O:C ratio needed for phase separation since they have used a limited number of systems and see exceptions to the ratio they state. and 2) the authors miss an opportunity to

hypothesize about the fundamental chemistry that underlies the phenomenon that they observe.

➔ To address the referee's comments we have 1) made it clearer in the revised manuscript that the average O:C is not the only parameter needed to predict LLPS (Abstract and page 6 line 7 – 8) and 2) included some discussion on the possible fundamental chemistry that underlies the phenomenon (page 8, line 15 - 28). Specifically, the following text have been added to the manuscript.

"Figure 5 also suggests that the average O:C is an important factor (although not the only factor) in determining the occurrence of LLPS in organic particles. For particles containing one organic species, LLPS was observed when the O:C was ≤ 0.44 (but not in all cases); for particles containing two organic species, LLPS was observed when the O:C was ≤ 0.58 (but not in all cases); and for particles containing SOM, LLPS was observed when the O:C was ≲0.44. In Fig. 5, the O:C range founded in ambient organic aerosols is also shown (Zhang et al., 2007; Hallquist et al., 2009; Jimenez et al., 2009; Heald et al., 2010; Ng et al., 2010). Based on this range and the range over which LLPS was observed in one-component organic particles, two-component organic particles, and SOM particles, LLPS is likely a common feature of organic aerosols in the atmosphere.
In addition to the average O:C and the types of functional groups, the spread in the O:C values of the organic molecules with the same particles is likely important for LLPS. Additional studies are needed to isolate the effect of average O:C, functional groups, and spread in O:C values on LLPS in organic particles."

*[2]* Table 1: It would be helpful to have the molecular structure of these species listed as well. Please also include solubility.

➔ As suggested, we have added molecular structure and solubility to Table 1 of the revised manuscript.

*[3]* Table 3: It would be helpful to know the mass fractions of the compounds in the mixture rather than having to back calculate based on the O:C ratio.

➔ All experiments for two organic species were conducted with the mass ratio of 1:1. This information has been added to Sect. 2.2 of the revised manuscript.

*[4]* pg 5 line 5: Can you comment on the "stems" that seem to connect the inclusions to the edge of the particles in Figs. 1b and 1c? I assume these are an optical artifact.

➔ Yes, it is an optical artifact in Figs. 1b and 1c. In the revised manuscript this has been mentioned in the figure caption for clarity.

*[5]* pg 5 line 8: How do you know that the organic-rich phase is the outer phase? Are you basing this on surface tensions or was it measured? I have the same question for pg 6 lines 10 and 15.

➔ We assume that the core of the particle is water-rich because the size of the core decreases and eventually disappears as the RH decreases. This assumption is also consistent with the surface tensions of organics and water, spreading coefficient theory,

aerosol optical tweezers experiments, and X-ray microscopy measurements (Jasper, 1972; Kwamena et al., 2010; Reid et al., 2011; Song et al., 2013; O'Brien et al., 2015; Gorkowski et al., 2016; Gorkowski et al., 2017). To address the referee's comment, we have added this information to the revised manuscript (Sect 3.1).

*Jasper, J. J.: The surface tension of pure liquid compounds, J. Phys. And Chem. Ref. Data, vol 1, 841-1009, Doi: http://dx.doi.org/10.1063/1.3253106, 1972.*

*Kwamena, N. O. A., Buajarern, J., and Reid, J. P.: Equilibrium Morphology of Mixed Organic/Inorganic/Aqueous Aerosol Droplets: Investigating the Effect of Relative Humidity and Surfactants, J. Phys. Chem. A, 114, 5787-5795, Doi 10.1021/Jp1003648, 2010.*

*Reid, J. P., Dennis-Smither, B. J., Kwamena, N. O. A., Miles, R. E. H., Hanford, K. L., and Homer, C. J.: The morphology of aerosol particles consisting of hydrophobic and hydrophilic phases: hydrocarbons, alcohols and fatty acids as the hydrophobic component, Phys. Chem. Chem. Phys., 13, 15559-15572, Doi 10.1039/C1cp21510h, 2011.*

*Song, M. J., Marcolli, C., Krieger, U. K., Lienhard, D. M., and Peter, T.: Morphologies of mixed organic/inorganic/aqueous aerosol droplets, Faraday Discuss., 165, 289-316, Doi 10.1039/C3fd00049d, 2013.*

*O'Brien, R. E., Wang, B. B., Kelly, S. T., Lundt, N., You, Y., Bertram, A. K., Leone, S. R., Laskin, A., and Gilles, M. K.: Liquid-Liquid Phase Separation in Aerosol Particles: Imaging at the Nanometer Scale, Environ. Sci. Technol., 49, 4995-5002, 10.1021/acs.est.5b00062, 2015.*

*Gorkowski, K., Beydoun, H., Aboff, M., Walker, J. S., Reid, J. P., and Sullivan, R. C.: Advanced aerosol optical tweezers chamber design to facilitate phase-separation and equilibration timescale experiments on complex droplets, Aerosol. Sci. Tech., 50, 1327-1341, 10.1080/02786826.2016.1224317, 2016.*

*Gorkowski, K., Donahue, N. M., and Sullivan, R. C.: Emulsified and Liquid Liquid Phase-Separated States of alpha-Pinene Secondary Organic Aerosol Determined Using Aerosol Optical Tweezers, Environ. Sci. Technol., 51, 12154-12163, 10.1021/acs.est.7b03250, 2017.*

*[6]* pg 5 line 14: Ciobanu et al. 2009 make a distinction between nucleation and growth from the edge of the particle vs. nucleation and growth that originates in the interior of the particle. It looks like Movies S4 and S6 show nucleation and growth that originates at the edge, and in Movie S5, the growth originates from the interior of the particle. If this result is consistent across many particles, it is worth pointing out to the reader.

➔ Thank you for the comment. Nucleation and growth begun from the edge of the particles of diethyl sebacate (Movie S4) and suberic monomethyl ester (Movie S6) while nucleation started from the interior of the particles of glyceryl tributyrate (Movie

S5). Such process was observed in all particles. This information has been added to the Sect. 3.1 of the revised manuscript.

*[7]* pg 5 line 22: Was there any size dependence to the range of LLPS values that is observed across a number of particles?

➔ We did not observe a dependence of LLPS on the particle size, although only a small range of sizes were explored. This information has been added to the Sect. 3.1 of the revised manuscript.

*[8]* pg 5 line 30: I don't think you can draw any conclusion about the O:C ratios at which LLPS is observed based on Table 2 because some systems phase separate and others do not at O:C < 0.44 and there are not a lot of systems studied. Can you make any conclusions based on the functional groups?

➔ To address the referee's comment, we have modified the sentence on page 6 line 7 - 8 to the following: "Table 2 also illustrates that LLPS can occur (but not always) in organic particles containing one organic species when the O:C ratio is ≤ 0.44."
In addition, we have also added the following discussion: "In addition to the O:C ratio, the types of organic functional groups present in the molecules are also likely important for LLPS (Song et al., 2012b), since different functional groups lead to different strengths of intermolecular forces with water." The following reference has also been added: "Song, M., Marcolli, C., Krieger, U. K., Zuend, A., and Peter, T.: Liquid-liquid phase separation in aerosol particles: Dependence on O:C, organic functionalities, and compositional complexity, Geophysical Research Letters, 39, 10.1029/2012gl052807, 2012."

*[9]* Movie S7 was not included in the zip file I downloaded. Movies S8-S12 were not readable by Quicktime (even though they have the same file extension as the other to these solutions, would the same phenomenon be observed or would ammonium sulfate increase the solubility of these compounds at high RH (salting in behavior)?

➔ For the issue of the movies, we have checked with Copernicus Publication regarding readability and have added movie S7. Regarding LLPS process when ammonium sulfate is added, the presence of ammonium sulfate would likely enhance LLPS process due to salting out effect (Marcolli and Krieger, 2006) but further studies are needed to verify this point.

*Marcolli, C., and Krieger, U. K.: Phase changes during hygroscopic cycles of mixed organic/inorganic model systems of tropospheric aerosols, J Phys Chem A, 110, 1881-1893, Doi 10.1021/Jp0556759, 2006.*

**Response to Referee #2 (Reviewer comments in black text)**
Summary:

The paper investigates the liquid-liquid phase separation (LLPS) in organic particles. LLPS was observed when the O:C ratios of the particles were ≤ 0.44 for single organic species and ≤ 0.58 for mixtures containing two organic compounds. Most of the LLPS can be observed at RH close to 100%. The authors demonstrate that the LLPS may have important implications on the CCN properties of organic particles. The manuscript is well written, and the results are clearly presented. I would like to recommend this manuscript to be published in Atmos. Chem. Phys. If my following concerns are fully addressed.

Concern.

*[10]* 1. Page 4, Line 8. The dimensions of the particles were measured around 30-80 μm. Depending on the particle generation methods, sizes of the particles used in the LLPS studies in the literature vary from 1 to 80 μm. 80 μm used in this study is at the higher end. Does particle size affect the results of LLPS?

> ➔ The resolution of the microscope used in the current experiments was roughly 1 μm. From experience, detection of LLPS with our microscope setup is the clearest when the size of the particles are roughly 30 - 80 μm, although smaller sizes are possible with optical microscopy. We did not observe a size dependence for the LLPS, although only a narrow range of sizes were investigated. This information has been added to Sect. 3.1 of the revised manuscript.

*[11]* 2. Page 5, Line 29. a. Have the authors tried nonanedioic acid (C9H16O4) and tetradecanedioic acid (C14H26O4)? These are the isomers of suberic acid monomethyl ester and diethyl sebacate with the same O:C ratio. One may expect different results with these isomers. For example, due to the stronger polarity and hydrogen bonding of acids as compared to esters, the acid compounds may have stronger interactions with water molecules, leading to higher SRHs. Or no LLPS may be observed. These effects may actually be helpful to explain why the authors did not observe LLPS with poly (propylene glycol) and PEG-400. Can you comment on whether/how the intermolecular interactions between organic compounds and water affect LLPS?

> ➔ Thank you for the comment. Nonanedioic acid and tetradecanedioic acid are solid at room temperature. In our studies we only used liquid organics species to ensure the mixtures were homogeneous prior to nebulization.

> ➔ Regarding intermolecular interactions, see response to comment *[8]* above.

*[12]* b. In contrast to the experiments with organic + ammonium sulfate conducted in the same group (e.g., Bertram et al.1 and You et al.2), no LLPS was observed in this study using the same organic compounds (i.e., polyethylene glycol, PEG-400, and diethyl-L-tartrate). I wonder

if the authors have any explanation regarding the different observations between these two studies.

➔ The difference observations between the two studies is likely due to salting out by ammonium sulfate (Marcolli and Krieger, 2006). We are currently working on a manuscript that will focus on this point.

Marcolli, C., and Krieger, U. K.: Phase changes during hygroscopic cycles of mixed organic/inorganic model systems of tropospheric aerosols, J. Phys. Chem. A, 110, 1881-1893, Doi 10.1021/Jp0556759, 2006.

[13] 3. Page 6, Line 2 and Table 3 What is the molar/mass ratio between the two organic compounds in the mixtures in Table 1? Is it 1:1? Have you tried any other mixing ratios? When nebulizing the mixture, how can the authors be sure that the composition of individual particle was the same as its original mixture?

➔ All experiments for two organic species were conducted with the mass ratio of 1:1. This information has been added to the Sect. 2.2 of the revised manuscript. All organic species used in this study are liquid at room temperature. Based on visual observations, the mixtures of two liquid organics were one phase prior to nebulizing as described in Sect. 2.2, hence a change on composition of the individual particles was not expected due to nebulizing the mixture. This information has been added to the revised manuscript.

[14] 4. Page 6, Line. a. It is interesting that adding PEG-400 which itself did not give LLPS, expanded the LLPS RH range of other compounds. Does it simply due to the change of O:C ratio? If not, please explain it in more detail.

➔ In addition to the average O:C, the spread in O:C values within an organic particle and the types of organic functional groups are likely also important. The manuscript has been modified to make these points clear (Sections 3.1).

[15] b. For polypropylene glycerol/PEG-400, no LLPS was observed in the single organic species scenarios. But mixing them together yielded LLPS with a low SRH (~75%). Have the authors tried any other mixtures of the following: diethyl sebacate, glyceryl tributyrate and suberic acid monomethyl ester? I wonder if the SRH and LLPS RH range will also change? Can you comment on this?

➔ We are not sure if we completely understand the referee's comments. The following binary mixtures related to the referee's comments were studied: diethyl sebacate + glyceryl tributyrate, diethyl sebacate + suberic acid monomethyl ester, and glyceryl tributyrate + suberic acid monomethyl ester (see Table 3). Is the referee suggesting

we try ternary mixtures?    We have not tried any ternary mixtures.

*[16]* 5. Page 8, Line 7. SOM is a bulk mixture. Why the results of SOM are similar to those from single organic components (i.e., LLPS occurred with O:C ratio ≤ 0.44; narrower LLPS RH range was observed)? Is it just a coincidence? It seems that O:C ratio solely cannot fully explain the results. Can the authors comment on this?

➔ First, the uncertainty of O:C of the SOM mixtures are large and thus we cannot determine based on our results if the SOM results are in better agreement with the studies using one organic species or two organic species.    Second, we agree that the O:C ratio cannot solely explain the results.    The manuscript has been revised to try and make this point clearer (Section 3.3).

Minor comments

*[17]* 1. Page 4, Line 10: "deposing". Should it be "depositing"?

➔ We have corrected "deposing" to "depositing" in the revised manuscript.

*[18]* 2. Page 17, Table 1: For diethyl-L-tartrate, I don't think there are ether group and carboxylic acid group.

➔ Thank you for the correction. Diethyl-L-tartrate is alcohol and ester group. We have corrected it in Table 1 of the revised manuscript.

Reference

(1) Bertram, A. K.; Martin, S. T.; Hanna, S. J.; Smith, M. L.; Bodsworth, A.; Chen, Q.; Kuwata, M.; Liu, A.; You, Y.; Zorn, S. R. Predicting the relative humidities of liquidliquid phase separation, efflorescence, and deliquescence of mixed particles of ammonium sulfate, organic material, and water using the organic-to-sulfate mass ratio of the particle and the oxygen-to-carbon elemental ratio of the organic component. Atmos. Chem. Phys. 2011, 11 (21), 10995–11006.

(2) You, Y.; Renbaum-Wolff, L.; Bertram, A. K. Liquid-liquid phase separation in particles containing organics mixed with ammonium sulfate, ammonium bisulfate, ammonium nitrate or sodium chloride. Atmos. Chem. Phys. 2013, 13 (23), 11723–11734.